# Diagnostic Value of Artificial Intelligence-Assisted Endoscopic Ultrasound for Pancreatic Cancer: A Systematic Review and Meta-Analysis

**DOI:** 10.3390/diagnostics12020309

**Published:** 2022-01-25

**Authors:** Elena Adriana Dumitrescu, Bogdan Silviu Ungureanu, Irina M. Cazacu, Lucian Mihai Florescu, Liliana Streba, Vlad M. Croitoru, Daniel Sur, Adina Croitoru, Adina Turcu-Stiolica, Cristian Virgil Lungulescu

**Affiliations:** 1Institute of Oncology, Prof. Dr. Alexandru Trestioreanu, Șoseaua Fundeni, 022328 Bucharest, Romania; elena.mateianu@gmail.com; 2Doctoral School, Carol Davila University of Medicine and Pharmacy, 020021 Bucharest, Romania; 3Department of Gastroenterology, University of Medicine and Pharmacy Craiova, 2 Petru Rares Str, 200349 Craiova, Romania; boboungureanu@gmail.com; 4Department of Oncology, Fundeni Clinical Institute, 258 Fundeni St, 022238 Bucharest, Romania; irina.cazacu89@gmail.com (I.M.C.); adina.croitoru09@yahoo.com (A.C.); 5Department of Radiology & Medical Imaging, University of Medicine and Pharmacy Craiova, 2-4 Petru Rares St, 200349 Craiova, Romania; lucian.florescu@umfcv.ro; 6Department of Oncology, University of Medicine and Pharmacy Craiova, 2 Petru Rares Str, 200349 Craiova, Romania; lilianastreba@gmail.com (L.S.); cristilungulescu@yahoo.com (C.V.L.); 711th Department of Medical Oncology, University of Medicine and Pharmacy Iuliu Hatieganu, 400012 Cluj-Napoca, Romania; 8Department of Pharmacoeconomics, University of Medicine and Pharmacy of Craiova, 2 Petru Rares Str, 200349 Craiova, Romania; adina.turcu@gmail.com

**Keywords:** artificial intelligence, deep learning, computer-aided diagnosis, pancreatic cancer, endoscopic ultrasound

## Abstract

We performed a meta-analysis of published data to investigate the diagnostic value of artificial intelligence for pancreatic cancer. Systematic research was conducted in the following databases: PubMed, Embase, and Web of Science to identify relevant studies up to October 2021. We extracted or calculated the number of true positives, false positives true negatives, and false negatives from the selected publications. In total, 10 studies, featuring 1871 patients, met our inclusion criteria. The risk of bias in the included studies was assessed using the QUADAS-2 tool. R and RevMan 5.4.1 software were used for calculations and statistical analysis. The studies included in the meta-analysis did not show an overall heterogeneity (I^2^ = 0%), and no significant differences were found from the subgroup analysis. The pooled diagnostic sensitivity and specificity were 0.92 (95% CI, 0.89–0.95) and 0.9 (95% CI, 0.83–0.94), respectively. The area under the summary receiver operating characteristics curve was 0.95, and the diagnostic odds ratio was 128.9 (95% CI, 71.2–233.8), indicating very good diagnostic accuracy for the detection of pancreatic cancer. Based on these promising preliminary results and further testing on a larger dataset, artificial intelligence-assisted endoscopic ultrasound could become an important tool for the computer-aided diagnosis of pancreatic cancer.

## 1. Introduction

Pancreatic cancer (PC) is one of the most lethal cancers because of its relative treatment resistance and rapid progression [1]. It is difficult to diagnose early-stage PC due to the lack of specific symptoms and the absence of auxiliary examination modalities with high sensitivity and specificity. More than half of the patients present with distant metastases at the time of diagnosis with PC [2]. Currently, the therapeutic options for PC are limited, and surgery is the only effective treatment, but less than 20% of patients have resectable tumors at diagnosis [2,3]. As a result, research into novel approaches continues, and it is acknowledged that the use of emerging technologies to aid in earlier diagnosis is one of the most promising areas for investigation. Recent advances in the field of artificial intelligence applied to the augmentation of imaging modalities are encouraging, and we must recognize the potential and fully utilize the opportunities for interdisciplinary research to improve the prognosis of patients with PC.

Artificial intelligence (AI) is a mathematical predicting technique that automates data learning and pattern recognition. Deep learning is an artificial intelligence algorithm and advanced type of machine learning method that employs neural networks [4]. Deep learning is capable of high-performance prediction. It is commonly used in AI algorithms and has been used in medical diagnosis [5,6]. Deep learning and other machine learning techniques are expected to have a significant impact on medical image diagnosis, but these techniques are currently underdeveloped [7]. The application of AI to clinical diagnostics was developed in the early 1980s, and computer-aided diagnosis (CAD) systems using deep learning have recently been used to assist doctors in improving the efficacy of various medical imaging data interpretation [8,9,10]. In the field of gastrointestinal (GI) tract endoscopy, AI has a wide range of applications, including the detection of colon polyps and the diagnosis or estimation of the invasion depth of GI tract cancers [11,12].

Artificial neural networks (ANNs) are primarily used in pancreatic cancer diagnosis for the task of assigning patients to small group classes based on measured features. Briefly, an ANN is a computerized model that simulates the information processing mechanisms of the human brain. An ANN is defined by the connections, numbers, and distribution within layers of neurons [13].

Traditional machine learning (ML) approaches such as support vector machine (SVM) are better suited for analyzing relatively modest-sized data sets with many variables. This makes it difficult to obtain appropriate samples of patients who have not yet developed cancer, which still limits the applicability of omics-based deep learning algorithms for early detection of pancreatic cancer [14].

Among the machine learning algorithms related to image feature extraction and classification, convolutional neural networks (CNNs) have been widely proven to be superior to traditional ML algorithms. These networks provide the flexibility to extract discriminative features from medical images while preserving their spatial structure and could be developed for region recognition and classification of images for pancreatic cancer detection [7].

The stage at diagnosis can determine the prognosis and treatment of pancreatic cancer patients [3]. Thus, it is imperative to find an accurate and reproducible way to detect methods of early and better detection of PC. To address this issue, this meta-analysis was intended to assess the applicability of artificial intelligence (AI) in the diagnosis of PC.

## 2. Materials and Methods

### 2.1. Search Strategy and Study Selection

The present study was conducted following the principles of the Preferred Reporting Items for a Systematic Review and Meta-analysis of Diagnostic Test Accuracy Studies (PRISMA-DTA) statement [15]. An a priori defined review protocol was registered for this meta-analysis (PROSPERO—Centre for Reviews and Dissemination University of York, York, UK), but a number was not assigned yet.

The systematic literature search was carried out by two investigators independently (I.M.C., V.M.C.) in the following databases: PubMed, Embase, and Web of Science. We searched for articles published up to October 2021 using the following keywords alone or in combination: endoscopic ultrasound, endosonography, artificial intelligence, deep learning, computer-aided diagnosis, machine learning, and pancreatic cancer. The data search was limited to studies written in English, with no other restrictions. In order to identify other potentially eligible publications, the references from the studies identified initially were reviewed manually. Study investigators were contacted by email in an attempt to obtain missing data when required.

### 2.2. Inclusion and Exclusion Criteria

We carefully selected articles that accurately described the application of artificial intelligence to EUS for the diagnosis of PC.

The studies included in this study were required to meet the following criteria: (1) original studies using artificial intelligence to analyze EUS data for the diagnosis of pancreatic cancer; (2) the final diagnosis was established by the histopathologic examination of the surgically resected specimen or EUS-FNA/FNB sample; (3) inclusion of sensitivity, specificity, diagnostic accuracy, or sufficient information to construct contingency tables; (4) the study clearly described the CAD algorithms and the process applied in PC diagnosis.

We excluded review articles, case reports, letter to editors, abstract-only texts, comments, and studies where it was not possible to retrieve data clearly reporting the diagnostic accuracy of AI-based models.

### 2.3. Data Extraction

Two authors (I.M.C., V.M.C.) extracted the study characteristics from each included study. Disagreements were resolved through discussion and consensus or by consulting a third member (C.V.L.) of the review team.

### 2.4. Quality Assessment of the Studies

Two reviewers (I.M.C., V.M.C.) appraised all the included studies by using a checklist based on the Quality Assessment of Diagnostic Accuracy Studies 2 (QUADAS-2) guidelines [16].

### 2.5. Statistical Methods

RevMan 5.4.1 software (The Cochrane Collaboration, 2020, London, United Kingdom) and mada R-package (R foundation, Vienna, Austria) were used to perform the diagnostic meta-analysis. Pooled sensitivity and specificity were assessed by plotting a summary receiver operating characteristic (SROC) curve to investigate the performance of AI in the diagnosis of pancreatic cancer using a bivariate random-effects model and a Bayesian approach. A high diagnostic efficacy was considered for a value higher than 0.75 for area under ROC curve (AUC) and partial AUC (using only the region where false positive rates of studies were actually observed, and then normalized to the whole space). The pooled diagnostic odds ratio (DOR) and its corresponding 95% confidence intervals (CIs) were also obtained to estimate the overall accuracy (a favorable test has DOR higher than 100). Higgins I^2^ was calculated to demonstrate the level of heterogeneity (a value greater than 50% was a significant indicator of substantial heterogeneity). The χ^2^ test was used to verify the null hypothesis that sensitivities and specificities were equal for all the included studies. A *p*-value less than 0.05 was considered statistically significant.

## 3. Results

### 3.1. Electronic Search Results and Study Characteristics

The process for selecting studies included in this meta-analysis is described in the flow diagram in Figure 1. A total of 83 potentially relevant records were identified initially with the aforementioned search strategy; no further articles were identified from the review of the reference lists. After screening the titles and abstracts and removing duplicates, 14 articles remained for full-text review. Four more articles were excluded after detailed assessment. In total, 10 studies, including 1871 patients, met our inclusion criteria [17,18,19,20,21,22,23,24,25,26]. The main characteristics of the studies are presented in Table 1. Norton et al. [19] published the first report of using CAD for pancreatic EUS in 2001. During the subsequent years, there were several reports of traditional CAD, which included computer-based extraction and selection of appropriate features that were further analyzed using a machine learning algorithm. Deep learning-based CAD was introduced in 2019 [18].

### 3.2. Quality of Included Studies

As shown in Figure 2, we found an unclear risk of bias in patient selection for Kuwahara 2019 [18], which was a retrospective study with no details on how the allocation list was conceived, and Zhang 2010 [25], where it was unclear whether patients’ randomization was performed. Das 2008 [17] was considered at high risk of bias for the index test because the index test results were not interpreted without knowledge of the results of the reference standard. Marya 2020 [20], Ozkan 2015 [21], Tonozuka 2021 [23], Udristoiu 2021 [24], and Zhang 2010 [25] were unclear about whether a threshold of the index test was used. Ozkan 2015 [21] was unclear about whether the reference standard results were interpreted without knowledge of the results of the index test and about whether there was an appropriate interval between index and reference test.

Das 2008 [17] and Marya 2020 [20] were considered having high risk of bias for flow and timing because not all patients received the same reference standard.

### 3.3. Diagnostic Accuracy

Pooled sensitivity and specificity, DOR with their 95% confidence intervals, and AUC with partial AUC are summarized in Table 2, with data stratified into several subgroups. All ten studies with 1871 patients were merged to derive pooled diagnostic test accuracy. The overall diagnostic accuracy showed 0.92 (95% CI, 0.89–0.95) sensitivity and 0.9 (95% CI, 0.83–0.94) specificity.

The most noticeable feature of the forest plot below (Figure 3) is the greater certainty of most of the studies (indicated by the confidence interval width).

The SROC curve in Figure 4 is shown by the black solid curve through the estimated mean (sensitivity, false positive rate): (0.92, 0.10). The same sensitivities were found between the studies (χ^2^ = 16.49, df = 9, *p* < 0.0574). Different specificities were found between the studies (χ^2^ = 57.84, df = 9, *p* < 0.0001). A high diagnostic efficacy was found with the AUC of 0.95. The partial AUC was 0.93. No heterogeneity between studies was found (Tau^2^ = 0.41, I^2^ = 0%, Cochran’s Q = 8.714, *p* = 0.464). DOR (95% CI) was 128.99 (71.17–233.81).

### 3.4. Subgroup Analysis

The studies included in the meta-analysis did not show an overall heterogeneity (I^2^ = 0%), and no significant differences were found from the subgroup analysis.

#### 3.4.1. Subgroup Analysis Based on the Type of Computer-Aided Diagnosis

If we analyze based on the type of computer-aided diagnosis, deep learning had a higher accuracy of diagnosis than conventional type in pancreatic cancer patients. Only three studies were included in the subgroup of deep learning-based CAD to derive pooled diagnostic test accuracy. A fixed-effects model was used as in Figure 5, where the SROC curve estimated mean (sensitivity, false positive rate) was (0.95, 0.10). No 95% prediction contour was drawn because of the small number of included studies. The pooled sensitivity was 0.95 (95% CI, 0.89–0.98). No differences were found between the studies’ sensitivities (χ^2^ = 1.332, *p* = 0.514). The pooled specificity was 0.90 (95% CI, 0.78–0.95). The same specificities were found between the studies (χ^2^ = 2.71, *p* = 0.258). A high diagnostic efficacy was found with the AUC of 0.97. The partial AUC (restricted to observed false positive rates and normalized) was 0.94. No significant heterogeneity between studies was found (Tau^2^ = 0.702, I^2^ = 0%, *p* = 0.398). DOR (95% CI) was 161.15 (36.98–702.27).

Seven studies were included in the subgroup of conventional CAD to derive pooled diagnostic test accuracy. A fixed-effects model was used as in Figure 6, where the SROC curve estimated mean (sensitivity, false positive rate) was (0.92, 0.09). The pooled sensitivity was 0.92 (95% CI, 0.87–0.95). Some differences were found between the studies’ sensitivities (χ^2^ = 14.04, *p* = 0.029). The pooled specificity was 0.91 (95% CI, 0.79–0.96). Different specificities were found between the included studies (χ^2^ = 55.812, *p* < 0.0001). A high diagnostic efficacy was found with the AUC of 0.95. The partial AUC (restricted to observed false positive rates and normalized) was 0.93. No significant heterogeneity between studies was found (Tau^2^ = 0.566, I^2^ = 1.98%, *p* = 0.41). DOR (95% CI) was 138.25 (64.98–294.14).

#### 3.4.2. Subgroup Analysis Based on the Algorithm of Artificial Intelligence

Comparing after the algorithm of AI, the best accuracy was obtained for ANN type. Three studies reporting data on 395 patients were included in the analysis, as in Figure 7. The SROC curve estimated mean (sensitivity, false positive rate): (0.93, 0.08). No 95% prediction contour was drawn because of the small number of included studies. Since no heterogeneity was identified in our meta-analysis (Tau^2^ = 0.141, I^2^ = 0%), a fixed-effects model was applied for the pooled analysis. The pooled sensitivity was 0.93 (95% CI, 0.78–0.98), with different values between the sensitivities of the three studies (χ^2^ = 12.42, *p-*value = 0.002). The pooled specificity was 0.92 (95% CI, 0.86–0.95), the specificities of the three studies not being significantly different (χ^2^ = 0.846, *p-*value = 0.655). A high AUC was estimated: 0.95, almost the same as the partial AUC (0.91).

Four studies reporting data on 872 patients were included in the analysis, as in Figure 8. The SROC curve estimated mean (sensitivity, false positive rate): (0.91, 0.13). Since no heterogeneity was identified in our meta-analysis (Tau^2^ = 0.214, I^2^ = 16.04%), a fixed-effects model was applied for the pooled analysis. The pooled sensitivity was 0.91 (95% CI, 0.88–0.94), with the same values between the sensitivities of the four studies (χ^2^ = 3.47, *p-*value = 0.325). The pooled specificity was 0.87 (95% CI, 0.83–0.90), the specificities of the four studies not being significantly different (χ^2^ = 2.66, *p-*value = 0.447). A high AUC was estimated: 0.94, almost the same as the partial AUC (0.81).

Two studies reporting data on 604 patients were included in the analysis, as in Figure 9. The SROC curve estimated mean (sensitivity, false positive rate): (0.93, 0.02). No 95% confidence contour or prediction contour was drawn because of the small number of included studies. Since no heterogeneity was identified in our meta-analysis (Tau^2^ = 1.783, I^2^ = 0%), a fixed-effects model was applied for the pooled analysis. The pooled sensitivity was 0.93 (95% CI, 0.89–0.96). The pooled specificity was 0.98 (95% CI, 0.85–0.99), the specificities of the two studies being significantly different (χ^2^ = 4.486, *p-*value = 0.034). A high AUC was estimated: 0.93, almost the same as the partial AUC (0.92).

## 4. Discussion

With great attention to the development of AI, there is an increasing interest regarding its application in detecting and diagnosing cancer. In this systematic review and meta-analysis, we found that AI algorithms applied to EUS imaging may be used for the diagnosis of PC with very good diagnostic accuracy, in terms of sensitivity and specificity. There are several modalities used for the diagnostic imaging of PC, including contrast-enhanced abdominal computed tomography (CT), magnetic resonance imaging (MRI), endoscopic ultrasound (EUS), transabdominal ultrasound (US), and positron emission tomography-CT. Tumor detection rates were reported to be higher using EUS than US or CT [27]. Furthermore, the diagnostic performance of EUS depends on the experience and technical abilities of the EUS-operator. Sometimes, even if the practitioner is an expert, inadequate detection or misdiagnosis of tumors have been reported [28].

There is a strong interest in the research community to use data science methods, such as AI, to assist professionals in the field of medicine in detecting visual abnormalities while minimizing both false positives and false negatives. Combining the knowledge in the field and the capability of AI introduces a new world of exploration into both screening and diagnosis of PC. Literature on the use of AI in the diagnosis of PC shows no significant variation in diagnostic accuracy [29]. In the present study, the variation in diagnostic accuracy was not wide, and we obtained a pooled sensitivity of 92% and a pooled specificity of 90%.

Heterogeneity, which is common in diagnostic meta-analyses, is the result of variations among the different included studies [30]. These variations mainly include differences in the study population, study design, interventions, and interpretations of results. In our case, the heterogeneity between studies was not significant, regarding the study design, they were eight retrospectives [17,18,19,20,21,24,25,26] and two prospective studies [22,23]. A heterogeneous element was the target used for image analysis—some of the studies used B-mode image analysis [18,20,23], from texture features [17,25,26], digital features [21], or Grey-scale pixels [18] analysis, while others made their image recognition with time-intensity curves parameters from contrast-enhanced EUS [20] or were multiparametric (B-mode, contrast, elastography) [25].

Regarding the computer-aided diagnosis (CAD), we found in our meta-analysis two models of CAD: conventional and deep learning. Conventional CAD is often regarded as the “shallow-layer learning method”, which is where large amounts of data are used by computer systems to learn how to carry out specific tasks such as speech recognition [31]. Traditional machine learning relies on pattern recognition or statistical methods and requires structured and historic data with prior knowledge of outcomes [32]. In the conventional CAD subgroup, seven studies were included [17,19,20,21,22,25,26].

On the other hand, only three studies were included in the subgroup of deep learning-based CAD [18,23,24]. As we expected, deep learning had a higher accuracy of diagnosis than conventional type in PC patients, and an impressive diagnostic efficacy was found with the AUC of 0.97. One of the most promising areas of innovation in medical imaging in the past decade has been the application of deep learning. Deep learning has the potential to impact the entire medical imaging workflow from image acquisition and image registration to interpretation [33,34]. Deep learning models are typically trained with the assumption that both the training and testing sets are collected from the same distribution of patients; thus, if models are developed in one relatively homogenous population, it may not generalize to the diverse patient populations or clinical environments in the real world. Moreover, this form of bias does not only take shape in terms of patient demographics but can even surface itself in details, such as which machine the medical image was captured on [32]. To sum up, there were homogeneous data in the three deep learning-based CAD studies, but the use of data was only from three institutions, and it may not generalize to the diverse population patients in the world.

There were four types of AI algorithms used to generate the automatic, real-time diagnosis of PC: artificial neural network (ANN), with three studies [17,21,22]; convolutional neural network (CNN), with four studies [18,20,23,24]; and support vector machine (SVM), with two studies [25,26]; and there was only one basic machine learning model, unclassified [19]. Different as they are, algorithms have advantages of their own.

An ANN is defined by the neuron connections, numbers, and distribution in layers. Each neuron of the middle layer has a transfer function associated, and the signal flows from the input neurons to the output neurons of the final layer. ANN is one of the most preferred artificial intelligence techniques that aims to create a system similar to the operations of the human brain by imitating it. A multi-layered, feed-forward perceptron was used in the system that classified cancerous and noncancerous pancreas.

CNN is a type of artificial neural network used in image recognition and processing that is specifically designed to process pixel data; CNNs are powerful image processors that use deep learning to perform both generative and descriptive tasks and that can obtain global and local image information directly from the convolution kernels [35].

In our meta-analysis, the best accuracy to detect pancreatic cancer was obtained by ANN model algorithms. Three studies reporting data on 395 patients used this AI model with a pooled sensitivity of 0.93 (95% CI, 0.78–0.98). The diagnostic efficacy for this subgroup was the same as the overall efficacy, with an AUC of 0.95. On the other hand, the CNN-based studies presented a similarly high AUC of 0.94, almost the same as the partial AUC (0.81) and the SVM-based studies with an estimated AUC of 0.93. Interestingly, CNNs were not the best in the detection of PC, and they were included in the deep learning models, which as previously stated had a higher diagnostic accuracy than the conventional type in PC patients.

Although the outcome of our research seems to bring light to the application of AI in detecting PC from EUS imaging, several common limitations and defects should be noted. First, only ten original research articles met the selection criteria, as there were not many studies about the diagnostic value of artificial intelligence-assisted EUS models in pancreas cancer patients. We were also unable to retrieve sufficient data for some studies. The relatively small number of studies analyzed, and their heterogeneity and mainly retrospective nature entail a significant risk of selection bias.

Second, various indicators of diagnostic performance were used in the studies. The value of TP, TN, FP, and FN at a specified threshold should at least be provided, but most studies did not give a threshold or explain the reason for choosing this threshold.

For the long-term survival of PC patients, it is essential to detect small tumors. The sensitivity of detection of pancreatic tumors 3 cm in diameter was reported to be 93% for EUS, which was higher than that of contrast-enhanced CT (53%) and MRI (67%) [36]. Sakamoto et al. [37] reported that EUS can detect pancreatic cancers of 2 cm or less with a sensitivity of 94.4%. In our analysis, only two studies provided information about the dimensions of pancreatic masses [18,22], with a median range of lesion diameter of 31 mm for PC in Saftoiu et al. and a minimal malignant cyst size of 1 mm in Kuwahara et al. We cannot assess the sensitivity of detections of small tumors in our meta-analysis.

Furthermore, another heterogeneous element was the comparison between pathologies of the pancreas and histopathological types of PC. Three studies compared pancreatic ductal adenocarcinoma (PDAC) with chronic pancreatitis (CP) [19,22,26], with similar sensitivities but different specificities. PDAC was also compared with normal pancreas, pancreatic neuroendocrine tumors (NETs) [24], and autoimmune pancreatitis [20]. Only one study compared other types of pancreatic cancer: intraductal papillary mucinous neoplasm IPMN [18]. To sum up, digital features of EUS images are different between different pathologies and malignancies of the pancreas, so assessing the sensitivity of detection for various types of pancreatic masses can be challenging to determine.

Last but not least, in the 10 included studies there was not even one external validation, which means testing the model with an out-of-sample dataset from one or more other centers. Most studies split the dataset from one center into a training set and tested randomly or according to different parameters [17,22,25,26]. The performance was evaluated by the test set, which should be called internal validation. Since the goal of validation is to investigate the performance within patients from a different population, it is necessary to obtain a new dataset from a distinct source. As a result, the model’s generalizability could not be assured in the absence of external validation, causing the results to be overestimated.

AI can constantly provide reliable performance in a short amount of time, with the potential to compensate for a human’s limited capability and by preventing human errors in clinical practice. Therefore, our analyzed EUS-CAD systems can work not only in assisting the training of beginners of EUS instead of an instructor but also in supporting experts. However, to evaluate the real diagnostic performance of AI-based CAD, future studies will require additional patients from multiple sites.

## 5. Conclusions

Artificial intelligence-assisted endoscopic ultrasound is a promising, reliable modality for the diagnosis of pancreatic cancer in patients with pancreatic mass lesions, with high accuracy. Deep learning models used as a clinical decision supporting system could significantly improve diagnostic accuracy in the detection of pancreatic masses. In this review and meta-analysis, we also put forward some existing problems of design and reporting that the algorithm developers should consider. Based on these promising preliminary results and further testing on a larger dataset, artificial intelligence-assisted endoscopic ultrasound could become an important tool for the computer-aided diagnosis of pancreatic cancer.

## Figures and Tables

**Figure 1 diagnostics-12-00309-f001:**
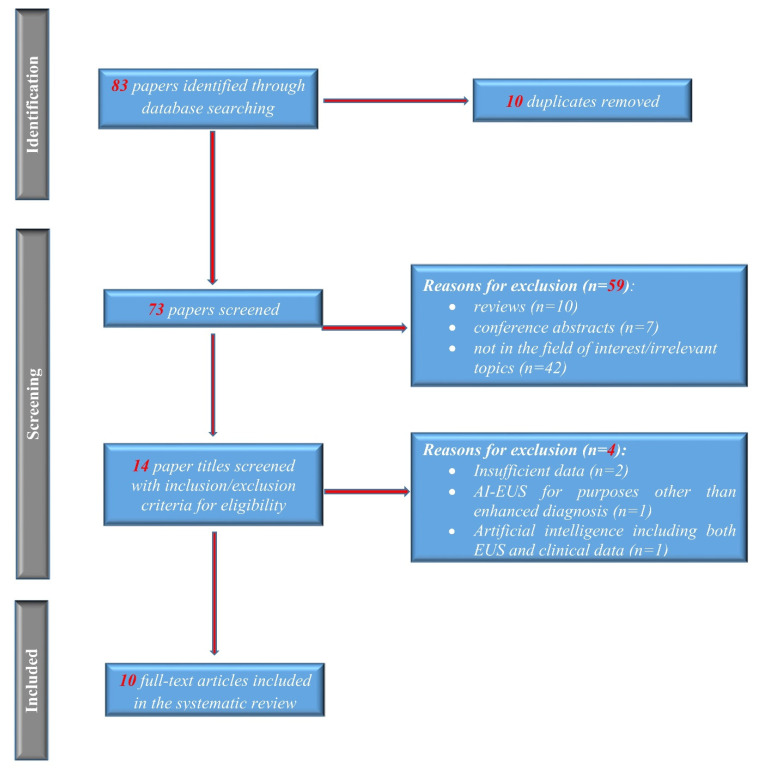
Study flow PRISMA diagram.

**Figure 2 diagnostics-12-00309-f002:**
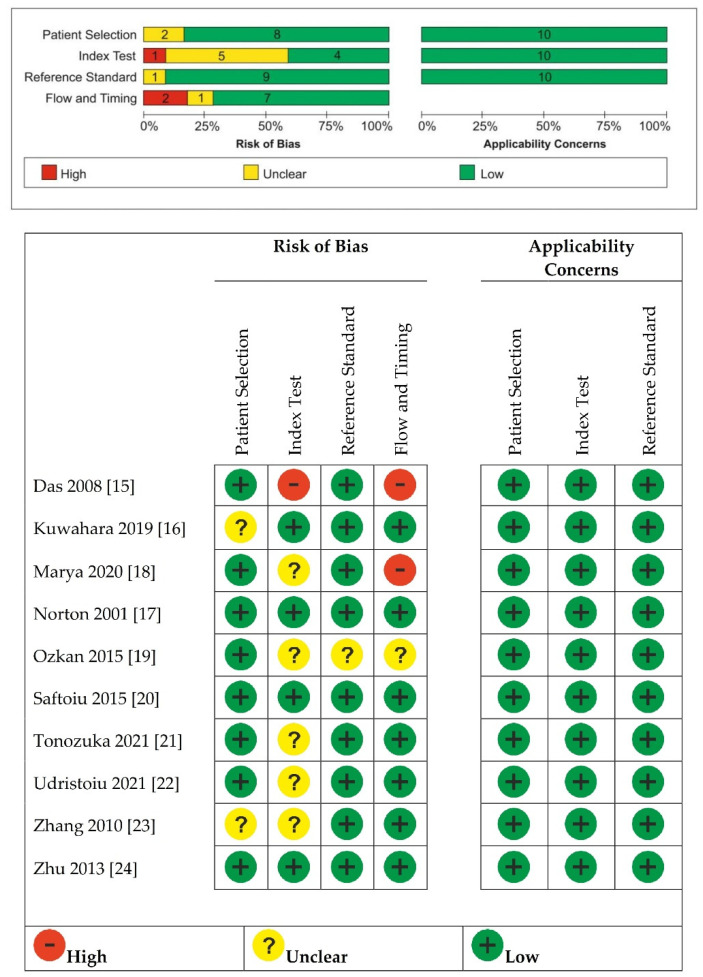
Quality assessment of included studies by using the QUADAS-2 assessment [15,16,17,18,19,20,21,22,23,24].

**Figure 3 diagnostics-12-00309-f003:**
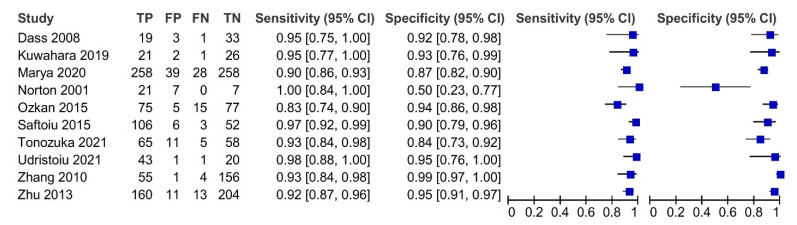
Forest plot with the diagnostic test accuracy (sensitivity, specificity, and 95% confidence interval) of each study for artificial intelligence in the diagnosis of pancreatic cancer.

**Figure 4 diagnostics-12-00309-f004:**
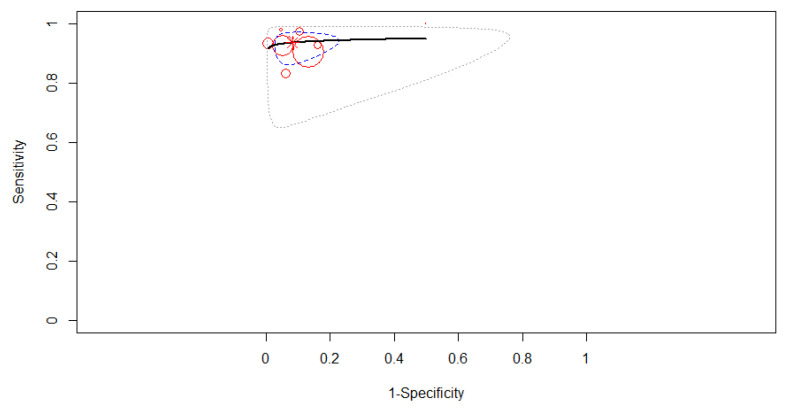
The SROC curve for AI. Dotted blue curve: 95% confidence region. Dotted closed curve: 95% prediction region for AI.

**Figure 5 diagnostics-12-00309-f005:**
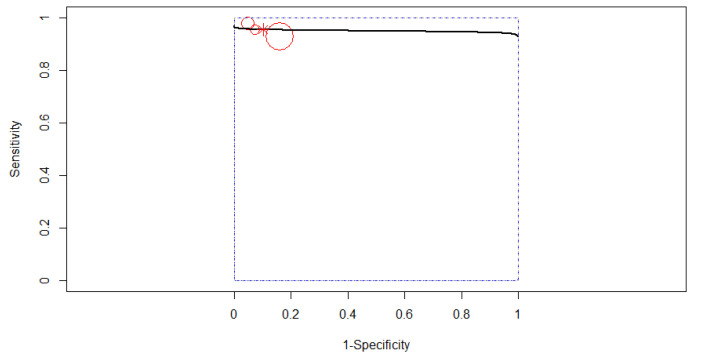
The SROC curve for deep learning-based computer-aided diagnosis.

**Figure 6 diagnostics-12-00309-f006:**
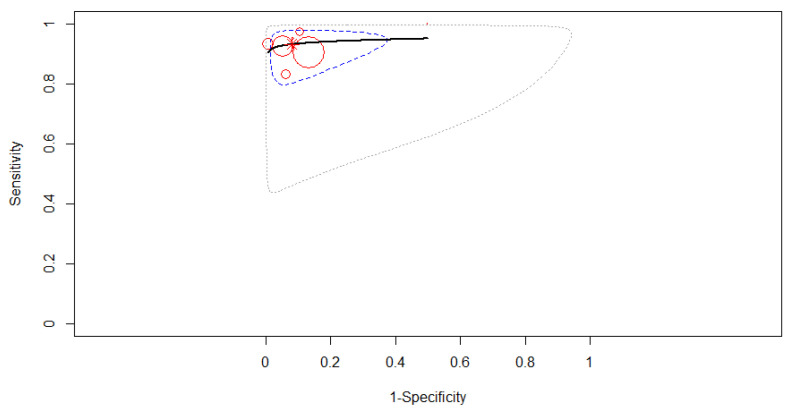
The SROC curve for conventional computer-aided diagnosis.

**Figure 7 diagnostics-12-00309-f007:**
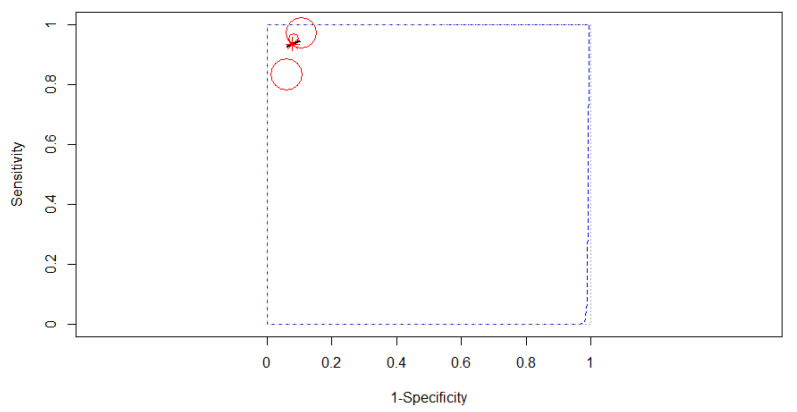
The SROC curve for ANN.

**Figure 8 diagnostics-12-00309-f008:**
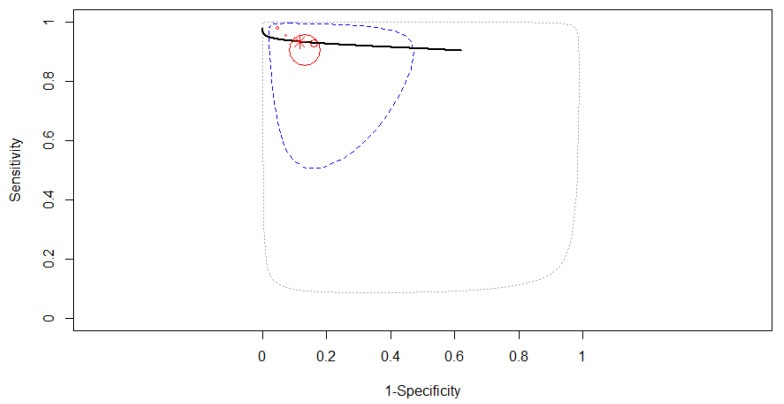
The SROC curve for CNN.

**Figure 9 diagnostics-12-00309-f009:**
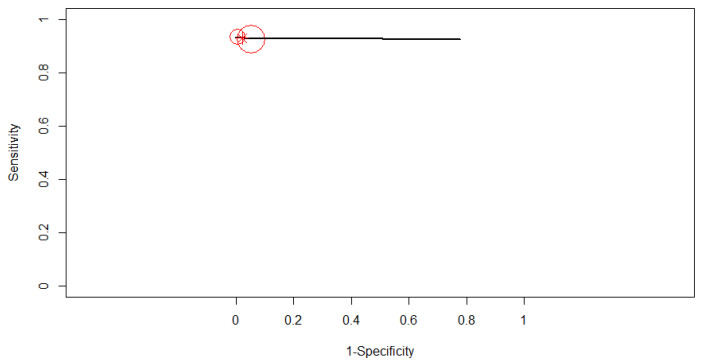
The SROC curve for SVM.

**Table 1 diagnostics-12-00309-t001:** The main characteristics of the included studies in the meta-analysis.

No. crt.	Author	Study Design	Comparison	No. of Patients (Overall Data)	No. of Images (Overall Data)	Testing Data	Final Diagnosis	Analysis Target	Type of Computer-Aided Diagnosis (CAD)	Algorithm of AI
1	Kuwahara 2019	retrospective	benign IPMN vs malignant IPMN	50	3970	no separate testing data	27 benign IPMN/23 malignant IPMN	B-mode images	deep learning-based CAD	CNN
2	Das 2008	retrospective	normal pancreas vs. chronic pancreatitis (CP) vs PDAC	56	319	50% of all data	2 normal pancreas/12 CP/22 PDAC	Texture features from B-mode image	conventional CAD	ANN
3	Marya 2020	retrospective	autoimmune pancreatitis vs. normal pancreas vs. CP vs. PDAC	583	1,174,461	123 patients	146 AIP/292 PDAC/72 CP/73NP	B-mode images	conventional CAD	CNN
4	Norton 2001	retrospective	CP vs PDAC	35	N/A	N/A	14 CP/21 PDAC	Grey-scale pixels from B-mode image	conventional CAD	Basic Neuronal Network/Machine Learning
5	Ozkan 2015	retrospective	PDAC vs. normal pancreas	172	332	72 (42 PDAC, 30 normal pancreas)	202 PDAC/130 normal pancreas (images)	Digital features from B-mode image	conventional CAD	ANN
6	Saftoiu 2015	Prospective	CP vs. PDAC	167		15% of pts	112 PDAC/55 CP	TIC parameters from contrast-enhanced EUS	conventional CAD	ANN
7	Tonozuka 2021	Prospective	normal pancreas vs. CP vs. PDAC	139	1390	47 pts, 470 images (25 PDAC, 12 CP, 10 NP)	76 PDAC/34 CP/29 normal pancreas	B-mode images	deep learning-based CAD	CNN
8	Udristoiu 2021	Retrospective	CP vs. PDAC vs. NET	65	3360	672 images from 65 pts	30 PDAC 20 CP/15 NET	Multi parametric (B-mode, contrast, elastography)	deep learning-based CAD	CNN
9	Zhang 2010	Retrospective	CP vs. PDAC vs. normal pancreas	216		50% of all data	153 PDAC/20 normal pancreas/43 CP	Texture features from B-mode image	conventional CAD	SVM
10	Zhu 2013	Retrospective	CP vs. PDAC	388		50% of all data (194; 131 PDAC, 63 CP)	262 PDAC/126 CP	Texture features from B-mode image	conventional CAD	SVM

**Table 2 diagnostics-12-00309-t002:** Pooled sensitivity, specificity, DOR, and AUC for AI.

	Number of Studies	Pooled Sensitivity (95% CI)	Pooled Specificity (95% CI)	Pooled DOR (95% CI)	AUC Partial AUC (Restricted to Observed FPRs and Normalized)
AI	10	0.92	0.9	128.9	0.95 (0.93)
		(0.89–0.95)	(0.83–0.94)	(71.2–233.8)	
CNN	4	0.91	0.87	86.2	0.94 (0.81)
		(0.88–0.94)	(0.83–0.9)	(39.7–187.2)	
ANN	3	0.93	0.92	141.5	0.95 (0.91)
		(0.78–0.98)	(0.86–0.95)	(55.8–358.9)	
SVM	2	0.93	0.98	547.9	0.93 (0.92)
		(0.89–0.96)	(0.85–0.99)	(64.3–4669.6)	
Deep Learning	3	0.95	0.9	161.2	0.97 (0.94)
		(0.89–0.98)	(0.78–0.95)	(36.9–702.3)	
Conventional	7	0.92	0.91	138.3	0.95 (0.93)
		(0.87–0.95)	(0.85–0.96)	(64.9–294.1)	

## Data Availability

The data that support the findings of this study are openly available.

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
