# Peer review of "Diagnostic Value of Artificial Intelligence-Assisted Endoscopic Ultrasound for Pancreatic Cancer: A Systematic Review and Meta-Analysis"

_diagnostics, 2022, doi:10.3390/diagnostics12020309_

Round 1
Reviewer 1 Report
This manuscript gave a systematic review and performed a meta-analysis of the existing research on AI--assisted endoscopic ultrasound for pancreatic cancer. 10 studies were finally selected and analyzed. The promising results demonstrated that AI technique could be an important tool for pancreatic cancer diagnosis.
Major comments:
- Chapter 2. Materials and Methods contains a little bit too many details about choosing related research, extracting data and results. Figure 1 plus some brief introductions should be enough.
- There’s no analysis among each subgroup that used similar technique. E.g., for the four methods used CNN framework, which of them performed better and which are the advantages of their method.
Minor comments:
- Please double check the typos in the manuscript, e.g.,
In. Table 2, should ‘0.86-95’ be ‘0.86-0.95’?
Author Response
We want to thank the reviewer for taking the time to revise our manuscript and for the on-point comments. We are confident that making the suggested modifications we will improve our manuscript.
Taking into account the prisma checklist we have to describe all of the required synthesis methods (http://prisma-statement.org/documents/PRISMA_2020_checklist.pdf). As suggested by the reviewer we have reduced some of the information where possible.
Furthermore, considering different algorithms were used in the ten included studies, even if no heterogeneity was found, we divided them into conventional and DL algorithms. In terms of algorithms of AI, there was ANN type (Das 2008, Ozkan 2015, Saftoiu 2015) with the best accuracy and among every subgroup, we analyzed the differences between the sensitivities and specificities of the studies: “The pooled sensitivity was 0.93 (95% CI, 0.78-0.98) with different values between the sensitivities of the three studies (χ2 = 12.42, p-value = 0.002). The pooled specificity was 0.92 (95% CI, 0.86-0.95), the specificities of the three studies being not significantly different (χ2 = 0.846, p-value = 0.655).”. These differences are shown in Figure 3, the forest plot with the diagnostic test accuracy.
In the case of CNN (Kuwahara 2019, Marya 2020, Tonozuka 2021, Udristoiu 2021), we analyzed the studies among this subgroup: “The pooled sensitivity was 0.91 (95% CI, 0.88-0.94) with the same values between the sensitivities of the four studies (χ2 = 3.47, p-value = 0.325). The pooled specificity was 0.87 (95% CI, 0.83-0.90), the specificities of the four studies being not significantly different (χ2 = 2.66, p-value = 0.447).” Looking also at Figure 3, we can observe none of them performed better inside the subgroup.
We have double checked the manuscript for typos.
Reviewer 2 Report
The review by Mateianu et al. entitled “Diagnostic value of artificial intelligence-assisted endoscopic ultrasound for pancreatic cancer: a systematic review and meta- analysis ” presents analysis of 73 papers on the pancreatic disease and AI. The final analysis on sensitivity and specificity of the results was done on 10 papers and the authors found relatively high results on these parameters. The authors concluded that artificial intelligence-assisted endoscopic ultrasound could become an important tool for the computer-aided diagnosis of pancreatic cancer. This is a well preformed meta-analysis and the results are significant to the topic of PC. The study may be accepted for publication after some small changes listed below.
- The abstract part is missing concluding remarks.
- In the introduction the authors could describe in more detail types of AI used for disease prediction.
Author Response
We want to thank the reviewer for taking the time to revise our manuscript and for the on-point comments. We are confident that making the suggested modifications we will improve our manuscript.
As suggested by the reviewer we have added the following paragraph to the abstract:
“Based on these promising preliminary results and further testing on a larger dataset, artificial intelligence-assisted endoscopic ultrasound could become an important tool for the computer-aided diagnosis of pancreatic cancer.”
We have added the following paragraphs to the introductions to describe in more detail the types of AI used:
“Artificial neural networks (ANNs) are primarily used in pancreatic cancer diagnosis for the task of assigning patients to small group classes based on measured features. Briefly, the ANN is a computerized model that simulates the information processing mechanisms of the human brain. An ANN is defined by the connections, numbers, and distribution within layers of neurons [13].
Traditional machine learning (ML) approaches like support vector machine (SVM) are better suited for analyzing relatively modest-sized data sets with many variables. This makes it difficult to obtain appropriate samples of patients who have not yet developed cancer, which still limits the applicability of omics-based deep learning algorithms for early detection of pancreatic cancer [14].
Among the machine learning algorithms related to image feature extraction and classification, convolutional neural networks (CNNs) have been widely proven to be su-perior to traditional ML algorithms. These networks provide the flexibility to extract dis-criminative features from medical images while preserving their spatial structure and could be developed for region recognition and classification of images for pancreatic cancer detection [7].”